# Validation of an Italian Questionnaire of Adherence to the Ketogenic Dietary Therapies: iKetoCheck

**DOI:** 10.3390/foods12173214

**Published:** 2023-08-26

**Authors:** Lenycia de Cassya Lopes Neri, Monica Guglielmetti, Valentina De Giorgis, Ludovica Pasca, Martina Paola Zanaboni, Claudia Trentani, Elena Ballante, Serena Grumi, Cinzia Ferraris, Anna Tagliabue

**Affiliations:** 1Faculty of Medicine, Department of Pediatrics, University of São Paulo, São Paulo 05403-000, Brazil; lenyciadecassya.lopesneri@unipv.it; 2Ketogenic Metabolic Therapy Laboratory, Department of Public Health, Experimental and Forensic Medicine, University of Pavia, 27100 Pavia, Italy; monica.guglielmetti@unipv.it (M.G.); claudia.trentani@unipv.it (C.T.); cinzia.ferraris@unipv.it (C.F.); anna.tagliabue@unipv.it (A.T.); 3Laboratory of Food Education and Sport Nutrition, Department of Public Health, Experimental and Forensic Medicine, University of Pavia, 27100 Pavia, Italy; serena.grumi@unipv.it; 4Child Neurology and Psychiatry Unit, IRCCS Mondino Foundation, 27100 Pavia, Italy; ludovica.pasca01@universitadipavia.it (L.P.); martinapaola.zanaboni@mondino.it (M.P.Z.); 5Department of Brain and Behavior Neuroscience, University of Pavia, 27100 Pavia, Italy; 6BioData Science Unit, Department of Political and Social Sciences, University of Pavia, Mondino Foundation, 27100 Pavia, Italy; elena.ballante@unipv.it

**Keywords:** adherence, patient compliance, ketogenic diet, epilepsy, GLUT1 deficiency, treatment adherence and compliance

## Abstract

Ketogenic dietary therapies (KDTs) are an effective and safe non-pharmacological treatment for drug-resistant epilepsy, but adherence can be challenging for both patients and caregivers. In Europe, there are no adequate tools to measure it other than monitoring ketosis. This study aimed to adapt and validate the Brazilian adherence questionnaire, Keto-check, into the Italian version: iKetoCheck. Using the Delphi technique, 12 judges validated the contents through agreement rates and the Content Validity Index (CVI). The iKetocheck was self-completed electronically by 61 drug-resistant epilepsy or GLUT1 deficiency patients within an interval of 15 days to measure its reproducibility. The test–retest reliability was evaluated using Pearson’s correlation and relative significance test. Exploratory and confirmatory factorial analyses were made using Factor software version 12.03.02. The final tool, iKetoCheck, consists of 10 questions with 5-point Likert scale answers. It evaluates various aspects such as informing caregivers about the diet, organization of meals, measurement of ketosis, weighing food consumed, diet negligence, use of carbohydrate-free medications, attending follow-up visits, reading food labels, consulting an expert for dietary concerns, and cooking at home. The factorial analysis resulted in three factors: “attention,” “organization,” and “precision,” with satisfactory results for indices in exploratory and confirmatory analyses. Although higher mean values of ketonemia measurement were observed in patients with a higher adherence score, these values were not statistically significant (*p* = 0.284). In conclusion, despite the small sample size, iKetoCheck is a valid tool for evaluating KDTs’ adherence in Italian drug-resistant epilepsy or GLUT1 deficiency patients. It can provide valuable information to improve patient management and optimize the effectiveness of KDTs.

## 1. Introduction

Ketogenic dietary therapies (KDTs) are effective and safe non-pharmacological treatments for drug-resistant epilepsy in both pediatric and adult patients [1]. KDTs are also considered the gold standard therapy for type I glucose transporter deficiency syndrome (GLUT1 DS) [2]. According to international guidelines, KDTs should be continued for at least three months to determine their effectiveness [3] and can be continued for years if effective [4].

There are different KDT protocols in use [5,6]: the Classic Ketogenic Diet (CKD), the Modified Ketogenic Diet (MKD), which is slightly different from the Modified Atkins Diet (MAD), the Medium Chain Triglyceride (MCT) diet, and Low Glycemic Index Therapy (LGIT). These hyperlipidic (65–90% of daily energy) and hypo-glucidic regimens induce a state of ketosis similar to the one that occurs after prolonged fasting. Medical nutritional monitoring is necessary to maintain their effectiveness over time and to reduce the risk of short- and long-term side effects [3].

Several hypotheses have been proposed to explain the mechanism of action of KDTs in controlling epilepsy, including molecular targets, mitochondria, neurotransmitter systems, and ketone bodies. However, no single theory has been identified [7].

CKD is the most commonly used treatment for GLUT1-DS and drug-resistant epilepsy, especially in infants, preschool children, and epileptic encephalopathies [3]. It is characterized by 80–90% of energy from fats, a normal intake of proteins (about 7–10% of energy, with a minimum of 1 g/kg/day), and a low content of carbohydrates (3–7% of energy). This treatment is rigorously individualized, minimally calculated, and based on the energy needs of each individual. All foods are weighed on a scale. Of all the protocols available, CKD promotes the highest levels of ketosis [8], but is also the most restrictive and less palatable [9], which can lead to lower adherence to the dietary treatment.

The World Health Organization (WHO) defines adherence as the degree to which a person’s behavior matches health care practitioners’ recommendations regarding medication, dieting, or lifestyle changes [10]. Compliance, concordance, and persistence are other terms generally used in the same context. However, according to some authors, they have different meanings. Jimmy and Jose (2011) referred to compliance as “the extent to which a patient’s behavior matches the prescriber’s advice and implies obedience to the physician’s authority” [11]. On the other hand, adherence is less paternalistic, since “patient and physician collaborate to improve the patient’s health” [11,12]. In agreement with this, the term concordance is introduced implying that the prescriber and the patient should come to an agreement about the regimen that the patient will take [13]. Persistence could be defined as “time from initiation of a chronic therapy to discontinuation of therapy” [14]. 

Long-term adherence to treatment with KDTs is often difficult for the patient and caregivers [5,15,16,17,18,19,20,21]. The results of a meta-analysis including 11 studies with KDTs in adults revealed combined compliance rates of 45%, 38%, and 56% for all types of KDTs, CKD, and an MAD, respectively [22]. Failure is mainly attributed to dietary side effects, psychosocial factors, the dietary restriction itself, as well as the diet’s inefficacy in seizure control [23,24]. Non-adherence to treatment (KDT and anti-seizure medication) is the main cause of treatment failure in patients with epilepsy and represents one of the biggest concerns for clinicians [5,15,16,17,18,19,20,21,25]. Non-adherence is also associated with health consequences, cost of visits, access to the ED, lack of seizure control, reduction in quality of life, etc. [26,27,28]. It is essential to educate the patient in adherence [29]. Patients with chronic pathologies are at risk of non-adherence; however, identifying patients with low adherence and the barriers to adherence appear fundamental for clinical outcome.

To date, adherence to KDTs has been evaluated both in the literature and clinical practice only through ketosis measurement (in urinary or blood samples). Usually, clinicians ask parents and patients to share the food diary to check the calorie intake, the ketogenic ratio, and the nutritional composition of the diet. A ‘Keto-check’ questionnaire was recently developed and validated in Brazil [30,31], aiming to provide a tool to measure adherence to the KDT. There are several advantages of using the questionnaire: low-cost, easy, fast, well accepted by the patient, valid and easily reproducible questions. This questionnaire is currently validated only in the Portuguese Brazilian language, and its items do not completely adapt to the Italian clinical practice. Thus, a re-arrangement of the Keto-check is necessary to create a quality tool validated by experts on a large population, allowing for a more solid assessment of adherence to treatment. Moreover, this instrument would facilitate Italian clinicians in interpreting efficacy results and implementing an early intervention to adjust therapy. Therefore, the objective of this study was to adapt and validate the Brazilian tool Keto-check into the Italian version: iKetoCheck.

## 2. Materials and Methods

This is a prospective observational study approved by the Ethical Committee (IRCCS San Matteo: number 0014551/22) and registered on ClinicalTrials.gov (NCT number: NCT05377762).

The process of validation was conducted in several phases: (i) the original questionnaire (Keto-check) [30,31] was initially adapted by the authors and underwent a Delphi technique for validation; (ii) reproducibility was tested in a sample of Italian patients; (iii) reliability was confirmed by factorial analysis; (iv) authors proposed a graphical representation of factors involved in adherence based on the factorial analysis.

### 2.1. Adaptation of the Brazilian Questionnaire

The questionnaire was initially translated from Portuguese into Italian by two registered dietitian specialists in KDTs, one being a native Portuguese speaker and the other a native Italian speaker. The translation was validated by a third-party Portuguese native speaker who subsequently arranged for the translation from Italian into Portuguese to verify the correctness of the previous translation.

The questionnaire consists of 10 questions relating to indicators about adherence to the ketogenic diet therapy, which include informing caregivers about the diet, organizing meals, measuring ketosis, weighing food consumed, diet negligence, using carbohydrate-free medications, attending follow-up visits, reading food labels, consulting an expert for dietary concerns, and cooking at home. This questionnaire can be completed directly by the patients or their caregivers, who will have to answer about the diet followed by the patient if the patient is a child, adolescent, or non-independent adult.

The patient/caregiver should indicate the level of agreement with the statement on a 5-point Likert scale: (1) strongly disagree, (2) partly disagree, (3) do not know how to answer, (4) partly agree, (5) strongly agree. Each answer was subsequently assigned a score and the sum of all the answers corresponded to the final score (minimum value 10, maximum 50). The higher the score, the better the adherence to the treatment.

### 2.2. Delphi Technique

To validate the tool, the modified Delphi technique was used [32]. The Delphi technique is a method structured in order to obtain a consensus between a panel of experts. Some parameters of this technique are not a standard among studies, for example the judge selection [33]. The tool was analyzed by a group of experts (members of the Italian Chapter of the International League against Epilepsy—LICE) acting as judges, composed of 12 professionals. Professionals were selected based on specific criteria and were invited to participate by email. They received a short text explaining the research and an informed consent form for participation. Next, the new KDT adherence verification tool was provided with questions regarding each statement used regarding relevance and ease of understanding. The answers were presented on a Likert scale with a score from 1 to 3: 1 = no; 2 = maybe; and 3 = yes. At the end of each question, a suggestion field was provided so that the judges could give feedback on the questions and enrich the content.

For the validation of the content, the agreement rate and the Content Validity Index (CVI) were calculated to represent the consensus among judges, as it measures agreement on each aspect assessed. The agreement rate was calculated by dividing the number of positive responses by the total number of responses. The CVI was calculated following four steps: (i) based on the judges’ scores (1 to 3), the average score for each entry was calculated. (ii) On the basis of the average, the initial CVI was calculated for each item by dividing it by the maximum value that the question could receive for relevance or clarity. (iii) The error of each entry was calculated to avoid any bias by the judges. The following formula was used to calculate the error: error = (1 ÷ number of judges) × number of judges. (iv) The final CVI of each element was calculated by subtracting the starting CVI from the error. The acceptable consensus rate in this study for each item analyzed was considered valid if it was greater than 90% or 0.9.

The validation of the content of the tool passed through two cycles, as necessary until the specialists reached the minimum value of the CVI for all questions. The answers obtained were organized in an Excel spreadsheet, with the numerical and subjective information compiled by the judges. After each cycle, the tool was reformulated based on the outcome of the discussion between the researchers. The modified instrument was sent again to all judges and discussed again until a consensus was reached.

### 2.3. Reproducibility

The questionnaire was completed electronically by patients via the Google Forms platform within a 15 day interval to measure its reproducibility. The test–retest reliability was evaluated using the Pearson correlation coefficient and relative significance test.

### 2.4. Study Population

#### 2.4.1. Judge Inclusion Criteria

Judges were selected based on the inclusion criteria outlined above and recruited from at least three national KDT application centers. The questionnaire was presented to judges in electronic form via the Google Forms platform to reach those who do not work in the same center. The inclusion criteria were: (1) to be a physician, dietician/nutritionist with a specialist, master’s, or PhD degree; (2) work in the fields of assistance, management, or teaching and research in KD therapy, with over 5 years of training and professional experience.

#### 2.4.2. Patient Inclusion Criteria

Patients were recruited voluntarily from the Center for Childhood and Adolescent Epileptology of the Mondino Foundation in Pavia and the Italian GLUT1 Family Association. At the time of enrollment, subjects were adequately informed about the project and the study objective. The privacy policy was provided and informed consent was obtained from the participants. For patients under 18 years of age, the questionnaire and related informed consent were provided by their parents or legal guardians. The inclusion criteria were: (i) KDT (all protocols) longer than 3 months; (ii) all genders; (iii) route of administration of the diet totally or partially oral; (iv) diagnosis of drug-resistant epilepsy or GLUT1 deficiency; (v) informed consent of the patient and/or authorized caregiver/legal representative consent. Exclusion criteria included: (i) KDT duration less than 3 months; (ii) route of administration of the diet exclusively enteral or parenteral diet; (iii) illiteracy.

### 2.5. Statistical Analysis

For the calculation of the sample size, a minimum of 5 subjects per item is recommended for the validation phase [34]. Taking into account the “rule of thumb,” a minimum of 100 participants (10 per item) was established. Considering a drop-out rate of 5%, a minimum of 105 patients were invited. For the reproducibility analysis, a subgroup of 50 subjects was considered representative as a minimum number, as recommended in the literature [35].

Reliability was verified through internal consistency using Cronbach’s Alpha coefficient, which can be classified as follows: values greater than 0.8 are considered “almost perfect,” 0.61 to 0.8 is “substantial,” 0.41 to 0.6 is “moderate,” 0.21 to 0.4 is “reasonable,” and less than 0.21 is “small” consistency [36]. These tests were performed using SPSS version 25, and the significance level was set at 5%.

To ensure data quality and control for potential random responses, data were excluded from the reproducibility analysis if the difference between the compilations of iKetoCheck by the same patient was greater than two standard deviations.

### 2.6. Factorial Analysis

Factorial analysis was conducted using Factor software [37] for both exploratory and confirmatory analyses. Optimal implementation of parallel analysis (PA) [38] was used to determine the number of dimensions in the exploratory analysis. All data were initially tabulated in an Excel spreadsheet and then transferred to Factor software [37] for parallel exploratory factor analysis [39]. Factor is a user-friendly program that is free and complete for conducting exploratory factor analysis and semi-confirmatory analysis models. It has been continually updated with recent proposals for factor analysis since its development more than 10 years ago [37].

#### 2.6.1. Exploratory Factor Analysis

Polychoric correlation was used to generate univariate descriptive results. This method is recommended when the univariate distributions of ordinal items are asymmetric or have excess kurtosis. Hot-deck multiple imputation was used to handle missing values in the exploratory factor analysis [40]. The adequacy of the polychoric correlation matrix was tested using Bartlett’s Test of Sphericity, where a significant value indicates that the correlation matrix is not an identity matrix and is suitable for factor analysis [41]. Kaiser-Meyer-Olkin (KMO) was also performed, where values above 0.6 are considered adequate for factor analysis [42].

Parallel exploratory factor analysis was conducted using a matrix of data collected and a matrix of data generated by chance through artificial intelligence tools. A sample of 500 data matrices (bootstrap samples with 95% confidence intervals) was generated to estimate the variance matrix. Optimal implementation of parallel analysis [43] was used to determine the number of dimensions. Robust Diagonally Weighted Least Squares (RDWLS) were used for factor extraction. The closeness to unidimensionality was assessed using three tests: unidimensional congruence (UniCo), where values larger than 0.95 are desirable; Explained Common Variance (ECV), where values larger than 0.85 are desirable; and Mean of Item Residual Absolute Loadings (MIREAL), where values lower than 0.300 suggest that the data can be treated as essentially unidimensional [44].

#### 2.6.2. Confirmatory Factor Analysis

After conducting exploratory factor analysis, confirmatory factor analysis was performed using the same sample, and the software Factor was used to check the robust goodness of fit statistics. The software performed all indices with a bootstrap of 95% confidence interval:Root Mean Square Error of Approximation (RMSEA), where values below 0.08 indicate good model fit [45].NCP (estimated non-centrality parameter), a measure used in statistical power analysis for hypothesis testing. It was adjusted with 26 degrees of freedom, and the test of approximate fit considered the null hypothesis RMSEA < 0.05 [46].NNFI (non-normed fit index), also known as the Tucker–Lewis Index (TLI), is a goodness-of-fit index used in structural equation modeling (SEM). The NNFI is a relative fit index that compares the fit of a hypothesized model to a baseline model. It is calculated as the difference in the chi-square values of the hypothesized model and the baseline model, divided by the degrees of freedom of the hypothesized model [47]. A NNFI value of 1 indicates perfect fit, while values closer to 0 indicate poorer fit.Comparative Fit Index (CFI), where values above 0.95 indicate good model fit [48].GFI (goodness of fit index), first proposed by Jöreskog and Sörbom (1981), generally, a GFI value of 0.90 or higher is considered indicative of good fit, although this threshold may vary depending on the specific research context [49].

## 3. Results

### 3.1. Adaptation of the Questionnaire

After the process of translation and back-translation (from Portuguese to Italian), the final questionnaire was approved as being similar to the Brazilian version. Following the creation of the final version (Appendix A), some adjustments were made to reflect the Italian ketogenic treatment reality based on the authors’ perceptions. Specifically, the first two questions were modified and the standard of answers (Likert scale) was simplified according to the frequency of behavior (ranging from “I never do” to “I always do”). 

### 3.2. Delphi Technique

Fifteen judges were invited but only twelve answered the first cycle; the general characteristics of the judges are in Table 1. In the second cycle, the same group of judges remained that answered before.

Table 2 presents all results of CVI and percentage of agreement between judges about relevance and clarity of each question from iKetoCheck on the first and second cycle of the Delphi technique.

### 3.3. Reproducibility

Characteristics of the sample are shown on Appendix A. The internal consistency of the full questionnaire showed a Cronbach’s Alpha of 0.704, which is considered substantial.

Twelve patients were excluded because their delta between compilations was greater than two standard deviations. The test–retest reliability, performed with 61 patients, showed an adequate Pearson’s correlation for each item, as displayed in Appendix A, while the test–retest reliability for the whole sample (*n* = 73) is shown in Appendix A.

### 3.4. Factor Analysis

The exploratory factor analysis resulted in three factors, as shown in the rotated loading matrix in Table 3. Bartlett’s statistic was 575.5 (df = 45, *p* = 0.00001) and the KMO test yielded a value of 0.684, indicating moderate sampling adequacy. The data were found to be not essentially unidimensional, with a UniCo of 0.841 (95% confidence interval 0.728–0.932), ECV of 0.695 (95% confidence interval 0.616–0.782), and MIREAL of 0.372 (95% confidence interval 0.289–0.432) [44]. The covariance matrix (polychoric correlation) of iKetocheck can be found in Appendix A.

The confirmatory factor analysis showed good results, with robust goodness-of-fit statistics after Losefer correction (RMSEA: 0.091, NCP *p* = 0.788; NNFI: 0.923, 95% confidence interval 0.845–0.952; CFI: 0.969, 95% confidence interval 0.938–0.98; GFI: 0.984, 95% confidence interval 0.950–0.990) [50].

The authors created a graphical presentation to illustrate the inter-relationships between factors and items, based on all the analyses performed (Figure 1). The graphic representation shows three factors and all the questionnaire items. The authors proposed the names of the factors as follows: factor 1, “attention,” which is mainly related to item 5 (avoiding eating foods not prescribed); factor 2, “organization,” which is mainly related to items 1, 2, and 7 (reporting the therapy to all family, preparing meals that allow you to follow the diet even when away from home, and attending appointments); and factor 3, “precision,” which is related to items 3 and 8 (measuring ketosis and filling out food and seizure diaries).

In order to compare with the Brazilian Keto-check, the same cut-off values were used, as can be seen on Table 4, notwithstanding this present questionnaire does not recommend any cut-off values. Although higher mean referred values of ketonemia measurement are observed in patients with a higher adherence score, these values are not statistically significant (ANOVA: F = 1.27, *p* = 0.28; Pearson correlation: 0.185), can be found in Appendix A.

## 4. Discussion

The KDT is considered the gold standard for the treatment of some rare neurometabolic diseases such as GLUT-1 deficiency and is a highly effective therapy for refractory epilepsy [2,3]. However, adequate adherence is essential to ensure its efficacy [3]. The validation of this questionnaire makes a practical non-invasive assessment of adherence possible, which could be applied periodically during interventions to monitor improvements in adherence to treatment.

The present study provides an innovative perception of KDT adherence. According to the analysis of grouped items by factors, three main factors were found: attention, organization, and precision. Attention could be correlated with the care of the basic informed therapy that allows the treatment to be considered adequate, with associated items such as avoiding not allowed food, reading food labels, and following instructions to weigh food. Organization permits a wider perception of the therapy, involving items such as informing all family members about the therapy, organizing meals outside the home, and attending all appointments. Precision is a factor that could make self-monitoring possible with ketosis assessment and monitoring diaries.

The concept of adherence is still unsettled in the scientific literature. Terms analogous are used as synonyms, such as compliance or concordance, despite some studies reinforcing an existing difference between these terms [12]. There is no gold standard tool for the assessment of adherence in KDTs. Some studies measure it by parental report of ongoing urinary or blood ketosis [51,52,53,54,55,56]. Some studies reported adherence as time on the diet [24,57]. Others report the reasons for discontinuation or duration of therapy and the difficulties referred to by patients [23,58,59,60,61,62,63,64,65,66,67]. Therefore, it is important to obtain a validated instrument to monitor adherence in a non-invasive manner. 

It is essential to identify patients at risk of low adherence, identifying barriers to adherence and the need for interventions aimed at increasing it [68]. Some caregivers have a positive perception and are convinced of the importance of KDTs, resulting in better adherence. Others are instead not sure about the necessity of this treatment and could not reach the same adherence level, with a reduced efficacy of the treatment [69]. 

Adherence plays a crucial role in clinical outcome and it is very important to consider adherence and develop specific interventions to improve it [68]. The literature reinforce some efforts to assist patients to adhere to the advice of health care providers, such as visual reminders, organizational strategies, behavioral strategies (e.g., simplification of the dosing regimen, the combination of step-by-step instructions with counseling, increasing follow-up, and using self-monitoring), and problem-solving around the adherence barrier (group motivational sessions and psychological therapy). The review also suggested that education, counseling, and behavioral interventions were effective strategies in patients with epilepsy [68]. Future studies are definitely needed to prove the questionnaire’s usefulness and adaptability to different sociodemographic and educational levels. Moreover, studies that correlate the questionnaire’s results with clinical parameters and efficacy data should be designed. A validation of the questionnaire in different languages deserves consideration since KDTs utilization and practice can vary among different countries, and thus specific modifications of the questionnaire’s items/questions might become necessary.

Some limitations of this paper must be considered. The sample size, for example, is one of the most important limitations which could have affected several aspects (e.g., representativeness). It is also possible that by recruiting a self-answered questionnaire sample, the results could be biased towards a more adherent population. Moreover, it is necessary to point out that the self-administration of the scale can imply random answers, affecting the validity of the results. However, in our study, random responses can be attributed to a small number of participants (*n* = 12) whose data were deleted to ensure adequate data quality, otherwise the analysis could be compromised by unreliable data. When these 12 compilations are included, a moderate impact on the test–retest correlation can be observed (Appendix A). To improve adherence, it is important to provide adequate parental counseling because KDT parent information is considered a key factor in maintaining long-term adherence and consequently plays a crucial role in clinical outcome, so it is very important to take adherence into consideration and develop specific interventions to improve it [20,68]. Another possible bias regards the representativeness of the sample. Although we disseminated the questionnaire through patient associations, we cannot be completely sure of having adequately characterized patients of different ethnicities, sociodemographic status, or other variables. Future studies might be addressed to broader populations and will mean adherence to clinical, educational, and sociodemographic data acquisition. 

Future prospects could involve developing psychoeducation courses to increase adherence. Therefore, it is important to create tools to measure KDTs efficacy and other aspects such as knowledge of the characteristics of the treatment and dedicated side effects questionnaires. 

## 5. Conclusions

The present study validated an indirect tool to measure adherence in KDTs in Italian patients. The interpretation of the factors related to adherence, named as attention, organization, and precision, enables the monitoring of care points during early intervention and implementation to adjust the therapy.

## Figures and Tables

**Figure 1 foods-12-03214-f001:**
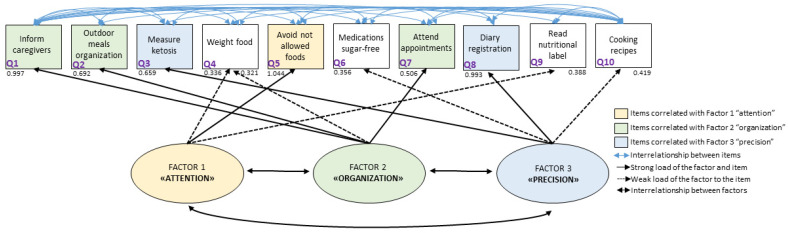
Graphical representation of 3 factors of adherence and related items.

**Table 1 foods-12-03214-t001:** Characterization of the judges who participated in the validation of the instrument’s content.

Characteristics	*n* (%)
*Gender*	
Female	10 (83.3)
Male	2 (16.7)
*Italian Region of professional activity*	
Lombardia	5 (41.7)
Veneto	3 (25)
Emilia Romagna	2 (16.7)
Friuli Venezia Giulia	1 (8.3)
Campania	1 (8.3)
*Area of expertise*	
Neurologist	7 (58.3)
Dietitian	3 (25)
Nutritionist (MD)	2 (16.7)
*Professional practice area*	
Assistance	2 (16.7)
Teaching and Research	0 (0)
Both	10 (83.3)

**Table 2 foods-12-03214-t002:** Results of Content Validity Index (CVI) and percentage of agreement between judges by Delphi technique of iKetoCheck for relevance and clarity, 2022.

Question		Relevance		Clarity	
		First Round	Second Round	First Round	Second Round
**1**	CVI	0.92	1	0.83	1
	% Agreement	83%	100%	67%	100%
**2**	CVI	0.94	1	0.83	1
	% Agreement	92%	100%	67%	100%
**3**	CVI	0.92	1	0.83	1
	% Agreement	83%	100%	67%	100%
**4**	CVI	0.94	1	0.86	1
	% Agreement	92%	100%	67%	100%
**5**	CVI	0.94	1	0.78	0.97
	% Agreement	92%	100%	50%	92%
**6**	CVI	0.94	1	0.86	0.94
	% Agreement	92%	100%	75%	92%
**7**	CVI	0.97	1	0.92	0.97
	% Agreement	92%	100%	83%	92%
**8**	CVI	0.97	1	0.89	0.97
	% Agreement	92%	100%	75%	92%
**9**	CVI	1.00	1	0.86	1
	% Agreement	100%	100%	67%	100%
**10**	CVI	0.94	0.97	0.89	0.97
	% Agreement	83%	92%	75%	92%
**Final Score**		**First round**		**Second round**	
	CVI	0.94		0.97	
	% Agreement	83%		92%	

**Table 3 foods-12-03214-t003:** Rotated loading matrix of questions of iKetoCheck.

Question	Factor 1	Factor 2	Factor 3
**q1**	−0.080	**0.997**	−0.050
**q2**	0.087	**0.692**	−0.045
**q3**	−0.207	0.293	**0.659**
**q4**	0.336	0.321	0.103
**q5**	**1.044**	−0.062	−0.078
**q6**	0.033	0.010	0.356
**q7**	0.341	**0.506**	−0.075
**q8**	0.078	−0.058	**0.993**
**q9**	0.388	0.164	0.194
**q10**	0.122	0.041	0.419

The bold indicates which factor is correlated for each question, some questions do not have a linkage with any factor.

**Table 4 foods-12-03214-t004:** Referred mean ketonemia values according Ketocheck’s categorization.

iKetocheck Adherence’s Categorization *	*n*	Mean	Standard Deviation	95% Confidence Interval
Insufficient	12	2.19	1.09	(1.50–2.88)
Good	56	2.38	0.91	(2.13–2.62)
Excellent	45	2.64	1.16	(2.29–2.99)
Total	114	2.46	1.04	(2.27–2.65)

* Considering scores equal or less than 35 as “insufficient”, from 36 to 44 as “good”, and higher than 45 as “excellent”.

## Data Availability

The iKetoCheck created can be found on Appendix A.

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
