# Peer review of "Validation of an Italian Questionnaire of Adherence to the Ketogenic Dietary Therapies: iKetoCheck"

_foods, 2023, doi:10.3390/foods12173214_

Round 1

Reviewer 1 Report

Dear editor and authors: The reviewed work has some strengths in terms of its rigor in the statistical treatment of the data, but it is really complex when it comes to reading it. I think they should present part of the results as Supplementary Tables and keep the really interesting results. For example, the sample description tables could become supplementary tables, and table 4, in which the correlations are placed diagonally, could well be cited in a single row. With this, it would allow a better focus on the importance of Table 5 and Figure 1.

Author Response

Thank you for your very useful suggestion. We moved Table 3 and 4 to the supplementary.

Reviewer 2 Report

This manuscript presents the validation of a Brazilian questionnaire to assess adherence to ketogenic diet therapy for epilepsy in an Italian population. This is a very relevant topic and the questionnaire could be of potential use in clinical and research environments.

Overall, the manuscript is very well-written and clearly presents the potential implications of the work.

Some comments below:

Introduction

The authors have not mentioned modified ketogenic diets (MKD – see Martin McGill 2019, ‘Understanding the core principles of a ‘modified ketogenic diet’: a UK and Ireland perspective’), which differs slightly to MAD. I recommend including this as a KDT type, potentially as an umbrella term including MAD.

The authors have missed a relevant study to include: Schoeler et al ‘Assessing parents' attitudes towards ketogenic dietary therapies’

‘Failure is mainly attributed to dietary side effects, psychosocial factors, or the dietary restriction itself [16].’ – what about lack of effectiveness, or waning effectiveness? This is often cited as the most common reason for stopping KDT.

From the ’Adaptation of the Brazilian questionnaire’ paragraph, it is implies that a healthcare professional guides the patient/parent when completing it (i.e. interviewer and interviewee), but the Reproducibility section says the questionnaire is completed on Google forms. This may be a slight confusion with English language, in particular with ‘This questionnaire can be addressed directly to the patients or to their caregivers…’, which is unclear. Please modify.

It appears that the Delphi technique adopted is not a truly traditional one. Perhaps refer to it as a ‘modified Delphi technique’ or similar, or provide further details in the Methods?

How can the authors be sure that the KDT population is adequately represented, in particular with regards different sociodemographic status / education level / ethnicity? If this was not done, please discuss as a limitation of the study.

Please provide a reference for the n=50 recommended for reproducibility analyses.

Results

‘Twelve patients were excluded because their delta between compilations was greater than 2 standard deviation’ – please explain this further and the impact it has on the results when they are included.

Discussion

The authors note their limited sample size but this should be emphasised further and noted in the abstract.

Please also discuss the twelve excluded patients and the implications on the interpretability of the results.

Could the authors indicate whether further study is needed? For example, to validate the questionnaire in English language?

Generally very good. Some minor alterations required - one of which is noted in the comments to Authors

Author Response

Thank you for your article´s revision. Please find all answers bellow:

Introduction

The authors have not mentioned modified ketogenic diets (MKD – see Martin McGill 2019, ‘Understanding the core principles of a ‘modified ketogenic diet’: a UK and Ireland perspective’), which differs slightly to MAD. I recommend including this as a KDT type, potentially as an umbrella term including MAD. Thank you, we included the MKD in the list of the KDT type.

The authors have missed a relevant study to include: Schoeler et al ‘Assessing parents' attitudes towards ketogenic dietary therapies’

Thank you for your precious consideration. We added this reference to the discussion (4th paragraph).

‘Failure is mainly attributed to dietary side effects, psychosocial factors, or the dietary restriction itself [16].’ – what about lack of effectiveness, or waning effectiveness? This is often cited as the most common reason for stopping KDT. Thank you, we added diet inefficacy as one of the reasons that may determine KDT interruption.

From the ’Adaptation of the Brazilian questionnaire’ paragraph, it is implies that a healthcare professional guides the patient/parent when completing it (i.e. interviewer and interviewee), but the Reproducibility section says the questionnaire is completed on Google forms. This may be a slight confusion with English language, in particular with ‘This questionnaire can be addressed directly to the patients or to their caregivers…’, which is unclear. Please modify. Thank you, we modified as reported “This questionnaire can be completed directly by the patients or their caregivers, who will have to answer about the diet followed by the patient if the patient is a child, adolescent, or non-independent adult. The patient/caregiver should indicate the level of agreement with the statement on a 5-point Likert scale

It appears that the Delphi technique adopted is not a truly traditional one. Perhaps refer to it as a ‘modified Delphi technique’ or similar, or provide further details in the Methods?

Thank you for your consideration. The word modified was added into the methods and some detail of this technique inserted into methods.

How can the authors be sure that the KDT population is adequately represented, in particular with regards different sociodemographic status / education level / ethnicity? If this was not done, please discuss as a limitation of the study.Thank you for pointing out this important element, we disseminated the questionnaire through the patients associations but we cannot be completely sure of the representativeness of the sample. Future studies might be addressed to broader populations and meant both to adherence, clinical, educational, and sociodemographic data acquisition.  We added it in the limitations

Please provide a reference for the n=50 recommended for reproducibility analyses.

Thank you, we have added the references in text (Kennedy, I. (2022). Sample size determination in test-retest and Cronbach alpha reliability estimates. British Journal of Contemporary Education, 2(1), 17-29.).

Results

‘Twelve patients were excluded because their delta between compilations was greater than 2 standard deviation’ – please explain this further and the impact it has on the results when they are included.

Thank you for the suggestion. As clarified in a previous comment, in this validation study the scales were online self-administered using google forms, implying a higher risk of random responses compared to the professional administration modality. This may reduce the reliability of the results and impact the analyses, therefore we identify a delta between the 2 compilations greater than 2 standard deviations as a possible index of random response. We have added this point also in the limitation section.

When these 12 compilations are included a moderated impact on the test-retest correlation can be observed, as shown in the supplementary material

Discussion

The authors note their limited sample size but this should be emphasised further and noted in the abstract.

Thank you for the suggestion, we modified the text accordingly.

Please also discuss the twelve excluded patients and the implications on the interpretability of the results.

We thank you for the considerations. We added into the discussion: “Moreover, it is necessary to point out that the self-administration of the scale can imply random answers, affecting the validity of the results. However, in our study, random responses can be attributed to a small number of participants (n=12) whose data were deleted to ensure adequate data quality, otherwise the analysis could be compromised by unreliable data.”

Could the authors indicate whether further study is needed? For example, to validate the questionnaire in English language?

We thanks the reviewer for this suggestion. Future studies are definitely needed to prove the questionnaire usefulness and adaptability to different sociodemographic and educational levels. Moreover studies that correlate questionnaire’s results with clinical parameters and efficacy data should be designed. A validation of the questionnaire in different languages deserves consideration since KDTs utilization and practice can vary among different countries and thus specific modifications of questionnaire’s items/ questions might become necessary.

Comments on the Quality of English Language

Generally very good. Some minor alterations required - one of which is noted in the comments to Authors

Thank you for all considerations.

Reviewer 3 Report

The present study validated an indirect tool to measure adherence in KDTs in Italian patients.

Interpretation of the factors related to adherence, named as attention, organization, and precision, enables monitoring of care points during an intervention and implementation of early intervention to adjust the therapy.

To improve adherence, it is important to provide adequate parental counseling because KDT parent information is considered a key factor in maintaining long-term adherence and consequently crucial role in clinical outcome, so it is very important to take adherence into consideration and develop specific interventions to improve it.

Future prospects could involve developing psychoeducation courses to increase adherence. Therefore it is important to create tools to measure KDTs efficacy and other aspects such as knowledge of the characteristics of the treatment and dedicated side effects questionnaires.

Author Response

Thank you for all kind considerations.
